# Iron Therapy in Chronic Kidney Disease: Days of Future Past

**DOI:** 10.3390/ijms22031008

**Published:** 2021-01-20

**Authors:** Kuo-Hua Lee, Yang Ho, Der-Cherng Tarng

**Affiliations:** 1Division of Nephrology, Department of Medicine, Taipei Veterans General Hospital, Taipei 11217, Taiwan; dadabim3520@gmail.com (K.-H.L.); yho@vghtpe.gov.tw (Y.H.); 2Institute of Clinical Medicine, National Yang-Ming University, Taipei 11217, Taiwan; 3Center for Intelligent Drug Systems and Smart Bio-devices (IDS^2^B), Hsinchu 300, Taiwan; 4Department of Biological Science and Technology, College of Biological Science and Technology, National Chiao Tung University, Hsinchu 300, Taiwan; 5Department and Institute of Physiology, National Yang-Ming University, Taipei 11217, Taiwan

**Keywords:** anemia, chronic kidney disease, iron therapy, ferric citrate, hypoxia-inducible factor

## Abstract

Anemia affects millions of patients with chronic kidney disease (CKD) and prompt iron supplementation can lead to reductions in the required dose of erythropoiesis-stimulating agents, thereby reducing medical costs. Oral and intravenous (IV) traditional iron preparations are considered far from ideal, primarily due to gastrointestinal intolerability and the potential risk of infusion reactions, respectively. Fortunately, the emergence of novel iron replacement therapies has engendered a paradigm shift in the treatment of iron deficiency anemia in patients with CKD. For example, oral ferric citrate is an efficacious and safe phosphate binder that increases iron stores to maintain hemoglobin levels. Additional benefits include reductions in fibroblast growth factor 23 levels and the activation of 1,25 dihydroxyvitamin D. The new-generation IV iron preparations ferumoxytol, iron isomaltoside 1000, and ferric carboxymaltose are characterized by a reduced risk of infusion reactions and are clinically well tolerated as a rapid high-dose infusion. In patients undergoing hemodialysis (HD), ferric pyrophosphate citrate (FPC) administered through dialysate enables the replacement of ongoing uremic and HD-related iron loss. FPC transports iron directly to transferrin, bypassing the reticuloendothelial system and avoiding iron sequestration. Moreover, this paper summarizes recent advancements of hypoxia-inducible factor prolyl hydroxylase inhibitors and future perspectives in renal anemia management.

## 1. Introduction

The World Health Organization defines anemia as a hemoglobin (Hgb) concentration of <13.0 g/dL in men and <12.0 g/dL in women [1]. Anemia is a common comorbidity in patients with chronic kidney disease (CKD), and its prevalence increases as CKD progresses to more advanced stages [2]. Anemia reduces the quality of life in patients with CKD and is associated with poor outcomes. Multiple mechanisms contribute to the development of anemia in patients with CKD. The primary factor leading to renal anemia is an inappropriately low erythropoietin (EPO) level in patients with CKD. Hence, the administration of erythropoiesis-stimulating agents (ESAs) has become the cornerstone of treatment for renal anemia [3]. However, studies have suggested that the use of ESAs to normalize Hgb levels in patients with CKD may lead to worse cardiovascular (CV) outcomes [4,5]. Hence, the Kidney Disease: Improving Global Outcomes (KDIGO) clinical practice guidelines for anemia in patients with CKD suggest setting the target Hgb level lower than the normal level [6].

In addition to relative EPO deficiency, iron deficiency is both common and one of the most critical causes of anemia in patients with CKD. Iron deficiency could occur as either a true paucity of iron stores (absolute iron deficiency) or as a relative deficiency (functional) in which the patient exhibits decreased iron utilization [7]. Iron supplementation is often necessary for treating renal anemia with iron deficiency. Iron supplementation therapy has taken on an increasingly important role in treating anemia in patients with CKD because of concerns regarding the safety and economic burden of using ESAs. Conventional oral iron preparations have demonstrated mixed efficacy in patients with CKD as well as gastrointestinal (GI) side effects. Intravenous (IV) iron supplementation is associated with anaphylactoid reactions. New iron supplement agents have recently emerged, such as ferric citrate (FC), which acts as an efficacious and safe phosphate binder that increases iron stores, thereby maintaining Hgb levels. New-generation IV iron supplements are also characterized by a diminished risk of infusion reactions. This paper reviews the physiology of iron metabolism, pathophysiology of renal anemia, and strategies of iron treatment, with a focus on novel iron supplement preparation and the safety and efficacy of these agents.

## 2. Normal Iron Metabolism

To maintain iron homeostasis, iron metabolism is strictly regulated by the interactions of numerous proteins (Figure 1).

Iron from dietary intake is absorbed by the duodenal enterocytes. The duodenal enterocytes reduce dietary ferric iron (Fe^3+^) to the ferrous state (Fe^2+^) by ferric reductase, duodenal cytochrome B (Dcytb), and then transport iron into the cell by divalent metal transporter 1 (DMT1). Iron is released into the circulation by export protein ferroportin (FPN) on the cell surface. The released of iron to the circulation is coupled to the oxidation of Fe^2+^ iron to Fe^3+^ iron, which is carried out by the iron oxidase hephaestin [8]. In the circulation, Fe^3+^ iron binds to plasma transferrin, the major plasma protein that transports iron. Some of the absorbed iron is then transported to the liver, taken up by hepatocyte through transferrin receptors (TfR1 and TfR2) and bound to ferritin for storage. The remainder of the absorbed iron is transported to bone marrow for erythropoiesis, a process stimulated by EPO, which is mainly secreted by renal pericytes. In erythropoiesis, iron deficiency limits the response to EPO [9]. At the end of the lifespan of red blood cells (RBCs), they are destroyed in the reticuloendothelial system (e.g., in the spleen), and the iron contained in the RBCs is recycled by macrophages. This process is referred to as iron recycling. Hepcidin, a peptide hormone mainly secreted by the liver, is recognized as the major regulator of iron homeostasis. High hepcidin concentrations downregulate both duodenal iron absorption and the release of iron from liver and spleen stores by downregulating cell surface FPN expression on absorptive enterocytes, macrophages, and hepatocytes [10]; low hepcidin concentrations upregulate iron absorption and iron recycling from macrophages. Hepcidin production is stimulated by increased iron uptake and inflammation and is suppressed under conditions of iron deficiency and hypoxia [11,12]. CKD is associated with elevated hepcidin levels [13,14]. Another key element involved in iron metabolism is hypoxia-inducible factor (HIF), a transcription factor that regulates erythropoiesis through an oxygen-sensing mechanism [15,16]. HIF comprises an oxygen-sensitive α-subunit (HIF-1α) and a stable β-subunit (HIF-1β). Under normoxic conditions, HIF-1α tends to be hydroxylated and eventually degraded by proteosome. Under hypoxic conditions, HIF-1α is stabilized and forms a dimer with HIF-1β, which then binds to hypoxia response elements (HREs) of the EPO gene and upregulates EPO transcription with the recruitment of coactivators such as p300 [17,18].

## 3. Pathophysiology of Renal Anemia

### 3.1. Relative Erythropoietin Deficiency

Multiple mechanisms contribute to the development of anemia in patients with CKD, with relative EPO deficiency being the most important factor. In patients without CKD, EPO levels rise in response to anemia. By contrast, in patients with CKD, EPO concentrations are inappropriately low for the degree of anemia [19]. Evidence suggests that reduced EPO production is associated with the transformation of peritubular fibroblasts into myofibroblasts [20]. Bernhardt and colleges demonstrated that EPO production could be increased in patients on dialysis through the inhibition of prolyl hydroxylase, which results in the stabilization of HIFs [21]. This suggests that a disturbed oxygen-sensing mechanism is also a key factor contributing to the development of renal anemia.

### 3.2. Shortened RBC Lifespan and Increased Blood Loss

Studies have demonstrated that RBC lifespan is reduced in patients with CKD [22]. Uremic RBCs have been reported to become more fragile in response to osmotic stimuli and oxidative stress [23,24]. Moreover, blood loss in patients with CKD is common due to dialysis and coagulopathy associated with CKD, which causes iron loss in these patients.

### 3.3. Chronic Inflammation and Iron Deficiency

CKD is often accompanied by chronic inflammation, which has a direct impact on erythropoiesis. Chronic inflammation also reduces the absorption and utilization of iron. In addition to absolute iron deficiency, many patients with CKD have functional iron deficiency, characterized by impaired iron release from body stores and impairment of iron utilization [25]. As mentioned, one of the key factors responsible for functional iron deficiency is inappropriately elevated hepcidin levels in patients with CKD, and elevated hepcidin levels are associated with chronic inflammation and reduced clearance of hepcidin in these patients [3].

### 3.4. Copper Deficiency

Copper is essential for absorbing iron from the gut. Copper is a crucial component of the ferroxidase enzymes, i.e., hephaestin and ceruloplasmin. Hephaestin is located in the duodenal mucosa and helps in the oxidation of ferrous iron, released from enterocyte FPN, to ferric iron. Ceruloplasmin is essential for iron transfer from macrophage to plasma. Balla and Ismail have reported that hemodialysis (HD) procedure could decrease serum copper levels in HD patients [26]. Therefore, copper deficiency may decrease ferroxidase activity, leading to impairment in iron absorption from the gut and subsequently Hgb synthesis [27].

### 3.5. Vitamin B12 and Folate Deficiency

Vitamins such as vitamin B12 and folate are necessary for the normal production of RBCs [28]. Many patients on dialysis are taking proton-pump inhibitors, known to be associated with subnormal vitamin B12 levels [29]. Meanwhile, high-flux HD is reported to lower vitamin B12 levels [30]. Vitamin B12 deficiency could cause the impairment of folate metabolism by the trapping of folate in the form of 5-methyltetrahydrofolate [31], leading to thymidine and purine synthesis impairment, resulting in ineffective erythropoiesis. Dialysis is also associated with a net loss of folate [32], leading to impaired DNA synthesis and ineffective erythropoiesis. However, folate deficiency is typically compensated by a regular diet or routine supplementation of water-soluble vitamins.

### 3.6. Aluminum Overload

Aluminum had been commonly used as a phosphate binder in patients on dialysis. Nowadays, the use of aluminum in dialysis-dependent patients has mostly been replaced by calcium-containing and non-calcium-containing phosphate binders. However, parenteral aluminum exposure via dialysate contamination is still observed [33]. Aluminum overload could lead to anemia and reduced ESA responsiveness by altered iron metabolism [34], direct inhibition of erythropoiesis [35], and disruption of the RBC membrane [36]. The improvement of anemia has been shown in patients using chelation therapy with desferrioxamine (DFO) [37].

## 4. Strategies for Iron Management in Patients with Chronic Kidney Disease and End-Stage Renal Disease

### 4.1. Absolute versus Relative (Functional) Iron Deficiency

Differentiating between absolute iron deficiency and relative (functional) iron deficiency is crucial. In cases of absolute iron deficiency, the total iron stores are depleted, limiting erythropoiesis. Absolute iron deficiency is generally caused by decreased GI absorption of iron and blood loss in patients with CKD. Patients with absolute iron deficiency typically present with a decreased iron level, decreased ferritin level, elevated total iron binding capacity (TIBC), and decreased transferrin saturation (TSAT; calculated as serum iron/TIBC × 100%). A TSAT of 20% is generally considered a threshold below which iron supplementation is indicated [38].

By contrast, relative (functional) iron deficiency occurs as a result of inefficient utilization of iron stores. Two main mechanisms contribute to the development of functional iron deficiency. One is chronic inflammation, and the other is the use of exogenous EPO, which results in increased production of RBCs. The available iron may be used faster than the existing iron stores are able to release it, leading to a supply/demand mismatch [11]. Patients with relative iron deficiency may have a TSAT of <20% because their bone marrow is stripping iron from the circulating transferrin faster than the transferrin can replenish it with iron released from stores [39]. Patients with relative iron deficiency may have a normal or elevated ferritin level. Notably, ferritin is an acute phase protein that exhibits elevated levels during infection and inflammation. Thus, elevated ferritin levels do not rule out the possibility of iron deficiency. Evidence suggests that ascorbic acid, a reducing agent, can improve iron utilization by facilitating the release of iron from ferritin and the mobilization of iron from the reticuloendothelial system to transferrin in patients with iron overload undergoing HD [40,41].

### 4.2. Optimal Target of Iron Supplementation

Iron therapy has advantages and disadvantages. Iron therapy in patients with CKD should be guided by iron status test results and clinical conditions. The decision to prescribe iron therapy should be made after its potential benefits have been weighed against its risks and harmful effects [42]. The benefits of iron therapy include avoiding or minimizing the need for blood transfusion and use of ESAs. The main risks and harmful effects of iron therapy include GI intolerability, anaphylactoid reaction, increased risk of tissue injury resulting from increased oxidative stress due to iron overload [43], and increased risk of infection. Currently, most guidelines suggest that iron supplementation therapy should be based on ferritin levels and TSAT. Serum ferritin concentration > 200 ng/mL and TSAT > 20% has been used as a target for patients on dialysis [44]. In 1996, Taiwan reached consensus on the diagnostic criteria for iron deficiency [45]. It is recommended that iron supplementation should be considered when a ferritin < 300 ng/mL and/or TSAT < 30% in dialysis patients and to maintain a ferritin level of 300–500 ng/mL and TSAT of 30–50%. The consensus was based on several previous studies performed in Taiwan and provided guidance on the use of IV iron to correct CKD anemia [46,47,48]. This recommendation on the management of anemia and iron deficiency in patients with CKD was years ahead of the current major CKD guidelines. Fishbane et al. [49] and Besarab et al. [50] have shown more reductions in ESA requirements by the use of IV iron supplementation to increase the ferritin to higher than 300 ng/mL and TSAT to 30–50%. In 2012, the KDIGO Anemia Management Guidelines recommended that for patients with anemia of CKD on dialysis, ESA treatment should be initiated when the Hgb concentration is between 9–10 g/dL to avoid having the fall of Hgb below 9.0 g/dL [6]. It is worthy of note that this recommendation has been complied with in Taiwan since 1996. In the Dialysis patients Response on IV Iron with Elevated ferritin (DRIVE) study, Coyne and colleges demonstrated that an intensive IV iron administration protocol (125 mg ferric gluconate for eight HD sessions) can significantly reduce ESA dosing requirements in patients with anemia on HD who have a ferritin ≥ 500 ng/mL and TSAT ≤ 25% and are receiving adequate epoetin [51]. Accordingly, in the KDIGO guidelines recommend using IV iron supplementation when the target is an increase in Hgb level or reduced dependency on ESA in patients with serum ferritin ≤ 500 ng/mL and TSAT ≤ 30% [6]. The most recent guidelines from the National Institute for Healthcare and Excellence (2015) [52] and the Renal Association (2017) [53] increased the ferritin ceiling to 800 ng/mL during iron supplementation therapy. A ferritin level of 800 ng/mL has been proposed as the threshold for withholding IV iron supplementation in patients on dialysis [54].

### 4.3. Iron Chelation Therapy for Iron Overload

CKD patients are vulnerable to iron overload due to the limited capacity of iron storage and transport via hepcidin modulation. As IV delivered iron exceeds the binding capacity of transferrin, the elevated plasma iron (LPI) contributes to the generation of reactive oxygen radicals that can trigger cellular damage [55]. Apart from the IV iron supplementation, frequent RBC transfusion could contribute to iron overload. The United States Renal Data System (USRDS) data have reported that blood transfusion is still widely used for anemia management in patients with CKD, particularly on dialysis, despite the widespread use of ESAs [56]. Previous studies have demonstrated that free iron may deposit in various tissues in persistent iron overload states, increasing the risks for liver cirrhosis [57], endocrine dysfunctions [58], diabetes [59], skin hyperpigmentation [60], cardiomyopathy [61], and immune dysfunction [62]. Accordingly, increased awareness of the risk of iron overload in CKD patients requiring chronic blood transfusion or high doses of IV iron is necessary.

Iron chelation therapy has been shown to reduce the iron burden and improve survival in patients with transfusion-dependent anemias [63,64]. These iron-chelating agents form iron complexes that promote iron excretion, reduce LPI levels, and remove excess iron from cells [65]. Nevertheless, few clinical trials have demonstrated the beneficial effects of iron chelators in CKD or HD patients with iron overload [37,66]. In clinical practice, the main challenges of iron chelation therapy may result from unfavorable pharmacokinetics and pharmacodynamics. For example, the rapid metabolism (30-min half-life) of DFO necessitates prolonged IV infusion for 12–15 h, leading to poor medication adherence. Besides, CKD patients may develop more significant adverse effects such as nephrotoxicity, hepatic failure, agranulocytosis, and teratogenicity in the treatment with deferasirox and deferiprone [67,68]. Fortunately, a recent experimental study reported a novel DFO-derived nanochelator exerting an improved efficacy and safety in treating mice models of iron overload. The renal clearable nanochelator can be administered subcutaneously and rapidly excreted by urine without demonstrated nephrotoxicity [69]. Compared to native DFO, this new compound may provide a safe and convenient option for iron chelation therapy in CKD patients suffering from iron overload in the future.

## 5. Current Advances in Oral Iron Supplementation

Ferrous salts are the first-line oral iron treatment, and ferrous sulfate is the most commonly used in clinical practice. Other conventional oral iron agents include ferrous gluconate, ferrous succinate, and iron polymaltose. The main problem with the clinical application of oral iron supplements is that most chronic diseases can lead to insufficient absorption and GI adverse reactions [70]. New oral iron agents that have emerged in recent years include FC, ferric maltol, heme iron polypeptide (HIP) and sucrosomial iron (SI). These modern iron formulations are associated with greater efficacy and fewer adverse effects than traditional iron preparations and are more effective at increasing Hgb levels [71]. Moreover, FC is an approved phosphate binder for use in patients with CKD who are dependent on dialysis. The availability of these new iron compounds broadens the possibilities of treatment of iron deficiency anemia (IDA) and enables better selection of iron preparations in various clinical situations, such as in cases of inflammatory bowel disease (IBD), celiac disease, and autoimmune gastritis. The following is a summary of the new preparations for oral iron supplementation (Table 1).

### 5.1. Ferric Citrate

FC was originally developed as a non-calcium phosphate binder, but in the phase 2 trial, researchers unexpectedly discovered that patients with non-dialysis-dependent CKD (NDD-CKD) and even patients with end-stage renal disease (ESRD) on dialysis could absorb most of the iron in the GI tract, leading to improvements in iron status and Hgb levels [72,73,74].

Since verification of the results of many phase 3 trials, FC has been approved by the United States Food and Drug Administration (FDA) for use in controlling serum phosphate levels in adult patients with CKD who are on dialysis and as an oral iron supplement for the treatment of IDA in patients with NDD-CKD [75,76,77]. Moreover, FC has also been reported to reduce the levels of C-terminal and intact fibroblast growth factor 23 (FGF23), which are strongly associated with CV diseases and are independent predictors of mortality in patients with moderate to severe CKD [76,77,78].

#### 5.1.1. Effects of Ferric Citrate on Phosphate Control

After oral administration of FC, the dissociated ferric iron binds to phosphorus in the GI tract and precipitates as ferric phosphate. This compound is excreted in the stool, thereby reducing dietary phosphate absorption. Several clinical trials have demonstrated the efficacy of FC as an iron-containing phosphate binder in reducing serum phosphate levels with various forms and stages of CKD [72,77,79,80,81,82,83]. For example, a prospective, double-blind, placebo-controlled, randomized trial conducted in five hospitals in Taiwan reported that FC effectively reduced serum phosphate levels compared with a placebo, with safety profiles similar to those of maintenance HD [83]. In our previous meta-analysis of nine randomized controlled trials (RCTs) including 1755 patients with CKD stages 3–5 and those on dialysis, we found that treatment with FC could effectively reduce serum phosphate levels by 1.39 mg/dL (95% confidence interval: 0.66 to 2.12) and had a comparable effect compared with active treatment, including calcium-based and non-calcium phosphate binders in phosphate control [84].

#### 5.1.2. Effects of Ferric Citrate on Iron Status and Hemoglobin Level

The intestinal absorption of iron from FC increases serum iron and ferritin levels and TSAT, thereby increasing the total Hgb level and RBC mass. Each 500 mg of FC (Nephoxil) contains 210 mg of elemental ferric iron, which is reduced to ferrous iron by DMT1 after oral administration [85]. After being transported into enterocytes through DMT1, iron can be stored in enterocytes bound to ferritin or exported into plasma through FPN and made available for erythropoiesis [86]. Fishbane et al. conducted a randomized, double-blind clinical trial on adults with NDD-CKD. Compared with individuals receiving a placebo, patients receiving FC were significantly more likely to exhibit increases in serum ferritin, TSAT, and Hgb level. The treatment effect was observed as early as 1–2 weeks after treatment initiation. The response was durable, resulting in significant reductions in the need for ESA and IV iron supplementation. Compared with traditional oral ferrous salts, the better GI tolerability of FC results from the lower generation of reactive oxygen species during iron metabolism [75]. A recent RCT also demonstrated that, in contrast to traditional oral ferrous iron supplements, which exhibit low potency for iron repletion, treatment with FC resulted in a more significant mean increase in TSAT and ferritin levels in patients with NDD-CKD [87]. Overall, in patients with moderate to severe CKD, oral FC could be a safe and efficacious treatment for IDA.

#### 5.1.3. Effects of Ferric Citrate on Fibroblast Growth Factor 23

Sustained hyperphosphatemia in patients with CKD typically results in an increase in FGF23, a hormone produced in osteocytes regulating phosphaturia. The production of FGF23 is stimulated by several factors, including altered levels of calcium, phosphate, parathyroid hormone, and 1-25-dihydroxyvitamin D [88]. Experimental data have also shown that inflammation and iron deficiency are both potent stimuli of FGF23 production [89,90,91,92]. In the early-stage of CKD, FGF23 acts directly on renal proximal tubules to induce phosphaturia, thereby reducing serum phosphate levels. However, persistently high concentrations of FGF23 are independently associated with left ventricular hypertrophy [93], anemia [94,95,96], and CV mortality [97,98]. In contrast to certain IV iron agents associated with elevated levels of FGF23, such as ferric carboxymaltose, iron polymaltose, and iron sucrose [99,100,101,102], FC has been reported to reduce FGF23 levels in patients with NDD-CKD and HD [103,104]. This reduction in FGF23 levels is also observed among individuals with normal baseline serum phosphate levels, suggesting that the effect is not mediated by reductions in serum phosphate levels. Possible mechanisms of this effect include attenuation of inflammation–FGF23 positive feedback loops and modification of the interactions between FGF23 and 1,25-dihydroxyvitamin D [76]. The clinical significance of reductions in FGF23 induced by FC requires further investigation. However, the resulting reduction of FGF23 has been speculated to increase 1,25-dihydroxyvitamin D levels and subsequently improve bone mineral density and cardiovascular functional capacity. Blocking FGF23 activity may also stimulate renal EPO production, alleviate renal anemia, and increase positive iron stores and iron utilization (Figure 2).

### 5.2. Ferric Maltol

Ferric maltol is a new iron complex consisting of a stable trivalent iron complex bound with three maltol ligands. The complex structure of ferric maltol remains intact before it is absorbed across the intestinal mucosa. This increases bioavailability and reduces GI toxicity associated with free iron. Additionally, maltol is believed to effectively inhibit iron-mediated lipid peroxidation and alleviate subsequent oxidative stress [105]. Oral ferric maltol is now approved in Europe and the United States for treatment in adult patients with IDA and IBD [106,107]. Moreover, preliminary data from a phase 3 trial revealed the non-inferiority of oral ferric maltol compared with IV ferric carboxymaltose administration in terms of Hgb response in the treatment of IDA in patients with IBD [108].

The Study with Oral Ferric Maltol for the Treatment of Iron Deficiency Anemia in Subjects With Chronic Kidney Disease (AEGIS-CKD) trial was a phase 3, randomized, placebo-controlled, multicenter study to investigate the efficacy of ferric maltol for treating patients with NDD-CKD. The pivotal results demonstrated the superiority of oral ferric maltol (30 mg twice daily) compared with a placebo in increasing Hgb levels after 16 weeks of treatment. Additionally, Hgb levels remained stable over a 36-week follow-up period for patients in the ferric maltol treatment group. The AEGIS-CKD study demonstrated that oral ferric maltol is a potentially safe and effective treatment that can benefit patients with CKD in the long term [71].

### 5.3. Heme Iron Polypeptide

HIP is a new-generation oral iron that uses the heme porphyrin ring to supply iron to absorption sites through a membrane protein, heme carrier protein 1, on the intestinal mucosa. In contrast to nonheme iron transported into the body through DMT1 affected by hepcidin, oral HIP can be better absorbed in patients with CKD and those with chronic inflammation [109].

Three RCTs were conducted to evaluate the bioavailability, tolerability, and efficacy of HIP in patients with CKD. Nagaraju et al. evaluated the effects of HIP on Hgb levels and iron profiles in adult patients with NDD-CKD. A total of 40 eligible participants were randomized to receive oral HIP or IV iron sucrose. No differences in Hgb level, TSAT, ESA dose, or adverse effects were observed between the two groups after six months of treatment, but ferritin levels were significantly higher in the IV iron sucrose group [110]. Thus, oral HIP has similar efficacy and safety to IV iron sucrose supplementation for maintaining Hgb levels.

Nissenson et al. conducted a prospective trial on patients on HD who had been receiving maintenance IV iron therapy to evaluate the effects of oral HIP supplementation as a replacement for IV iron therapy. After the six-month study period, the authors concluded that oral HIP treatment could successfully replace IV iron therapy in most of the patients on HD and maintain adequate Hgb levels without concomitant IV iron therapy. This treatment was associated with a significant increase in ESA efficiency [111]. These results suggest that oral HIP administration may be a reasonable option for iron supplementation in patients on HD receiving ESA therapy.

The HEMATOCRIT trial was a multicenter open-label, randomized controlled trial comparing the efficacy of HIP with oral nonheme iron for managing anemia in patients undergoing peritoneal dialysis (PD). They randomized 62 eligible patients on PD to receive either HIP or ferrous sulfate for six months. Although HIP has a significantly higher cost than nonheme iron, it was found to have no apparent benefits regarding safety or efficacy in patients on PD in terms of ferritin level and TSAT [112].

In summary, although early clinical data in healthy individuals suggest that orally administered HIP has superior bioavailability and tolerability compared with nonheme iron, current evidence shows that the effects of HIP administration on Hgb level, TSAT, and required EPO dose in patients with CKD who have anemia are not significantly different from those of IV iron or orally administered nonheme iron. In contrast to traditional iron agents, HIP is associated with low ferritin levels. Moreover, the cost of HIP is considerably higher than that of nonheme iron agents. A more extensive study must be conducted before HIP is widely adopted for iron supplementation in patients with CKD.

### 5.4. Sucrosomial Iron

SI is an innovative oral iron formulation in which ferric pyrophosphate is protected by a phospholipid bilayer and a sucrosomial shell. The unique structure enables iron absorption through various means (endocytosis, the paracellular pathway, and the M cells of Peyer’s patches) independent of DMT1 on the enterocyte surface [113]. The specific route of absorption in M cells causes iron to enter the lymphatic system instead of the blood. The high bioavailability of this preparation is likely due to bypassing of the conventional iron absorption pathway [114]. The safety and favorable tolerance of SI have been investigated in patients with various conditions, including CKD [115], solid tumors [116], IBD [117], and celiac disease [118]. Pisani et al. conducted an open-label RCT on patients with NDD-CKD to compare SI with IV ferrous gluconate administration in terms of the ability to ameliorate anemia. The results indicated that both could increase Hgb levels, but the increase was greater for IV iron administration. After the cessation of supplementation, the Hgb levels in the group receiving IV iron supplementation remained stable, but those in the SI group returned to the baseline level. IV iron administration demonstrated a greater effect on the repletion of iron stores, but the SI group had fewer adverse events [119]. Evidence is lacking regarding the long-term efficacy and safety of SI administration for the treatment of anemia in patients with CKD.

## 6. Current Advances in Intravenous Iron Supplementation

IV iron administration is appropriate for patients who are intolerant to oral iron therapy or who are undergoing HD. Numerous IV iron complexes are available, including iron sucrose, low- and high-molecular weight dextrans, and sodium ferric gluconate. However, the use of these traditional IV iron preparations requires attention to infusion reactions. The patient must be monitored for 60 min after the IV infusion to check for an allergic reaction. Moreover, the testing dose is recommended for the administration of traditional IV iron agents for the first time [120].

Recently, three new-generation IV iron compounds (ferumoxytol, iron isomaltoside 1000, and ferric carboxymaltose) were developed and began to be widely used in clinical practice. The most valuable characteristic of these agents is the high stability of the carbohydrate shell, which prevents uncontrolled release of toxic free iron, enabling complete replacement doses in one or two IV iron infusions. Their low immunological activity allows these preparations to be quickly administered as a single infusion without the need for a testing dose [121]. Another essential feature is the redox potential, defined as the substance’s propensity to be reduced, thereby generating reactive oxygen species after exposure to reducing agents such as excess ascorbic acid. The polynuclear iron core in these new IV iron agents is stable and has a low redox potential, thus minimizing the risk of oxidative stress reactions [122,123].

To prevent hepcidin-induced iron sequestration and oxidative stress after IV iron infusion, ferric pyrophosphate citrate (FPC) is an ideal option approved for maintenance iron therapy in adult patients on HD [79]. FPC is a highly water-soluble compound and is designed to be added to the bicarbonate concentrate used in every HD treatment. During each HD session, FPC can deliver approximately 7 mg of iron through dialysate [124]. The relevant treatment strategy is to provide small doses of iron that are directly transferred to transferrin, thus bypassing the reticuloendothelial system and limiting the formation of non-transferrin-bound iron (NTBI), which has the potential to induce the generation of reactive oxygen species and cellular damage [125]. In both healthy individuals and patients on HD, FPC administration reportedly does not cause increases in NTBI or markers of oxidative stress [124,126]. In two multicenter, randomized, placebo-controlled, phase 3 clinical trials, The Continuous Replacement Using Iron Soluble Equivalents (CRUISE 1 and 2), intradialytic iron supplementation using FPC was demonstrated to safely maintain Hgb levels and significantly reduce the required ESA dose in patients undergoing long-term HD [127,128]. A summary of modern IV iron formulations is provided in Table 1.

## 7. The Rising Star: Hypoxia-Inducible Factor Prolyl Hydroxylase Inhibitors

### 7.1. Mechanism of Action of Hypoxia-Inducible Factor Prolyl Hydroxylase Inhibitors

The production of EPO mainly depends on an oxygen-sensing mechanism performing feedback control, which is mainly based on the regulation of HIF-1. HIF-1 is a heterodimeric transcription factor consisting of a constitutively expressed β-subunit and an oxygen-regulated α-subunit. Under normoxic conditions, HIF-1α is hydroxylated by prolyl hydroxylase domain protein and subsequently recognized by the von Hippel–Lindau tumor-suppressor protein, a component of an E3 ubiquitin ligase complex. This interaction promotes the rapid degradation of HIF-1α. Under hypoxic conditions, hydroxylation does not occur and HIF-1α is stabilized. Consequently, the dimerization of HIF-1α with HIF-1β occurs to form HIF transcription factors, thereby promoting the transcription of target genes, including the EPO gene [129]. On this theoretical basis, HIF prolyl hydroxylase (HIF-PH) inhibitors, also known as HIF stabilizers, have been introduced for treating renal anemia, primarily through the stimulation of endogenous EPO production [130]. Without the use of ESAs, the administration of HIF-PH inhibitors can increase Hgb levels in patients with CKD, which implies that the pathogenesis of renal anemia is not only caused by insufficient EPO production but also the defective translation of the EPO gene [131]. Additionally, increased erythropoiesis was still observed in a nephrectomized rodent model after the use of HIF-PH inhibitors, which suggested that EPO-producing cells could also be located outside of the kidney [132,133]. HIF-PH inhibitors have multiple potential advantages over ESA treatments, including their ability to be administered orally and the fact that HIF-PH inhibition in animal models may also activate several target genes other than the EPO gene (e.g., the EPO receptor, transferrin, the transferrin receptor, ferroprotein, and DMT1), thereby stimulating the absorption of iron by the GI tract, the mobilization from macrophages into the circulation, and transportation the transferrin-bound iron to bone marrow for erythropoiesis [133]. More importantly, HIF-PH inhibitors may protect against the negative effects of inflammation on Hgb synthesis and RBC production by downregulating hepcidin [134,135].

### 7.2. Clinical Trials of Hypoxia-Inducible Factor Prolyl Hydroxylase Inhibitors

Clinical trials of HIF-PH inhibitors are being conducted in multiple countries worldwide. Among them, roxadustat, vadadustat, and daprodustat have all entered phase 3. The first published phase 3 study was a double-blind study that enrolled 154 Chinese patients with NDD-CKD who were randomly assigned to receive roxadustat or a placebo for an eight-week period. Parenteral iron therapy was withheld except as rescue therapy, which included blood transfusion, IV iron, and use of ESAs in the presence of significant anemia. The results indicated that roxadustat treatment increased Hgb levels by 1.9 ± 1.2 g/dL, whereas a decrease of 0.4 ± 0.8 g/dL was observed in the placebo group. Moreover, a greater reduction of hepcidin levels was observed in the roxadustat group than in the placebo group. Compared to the placebo group, treatment with roxadustat reduced the requirement of rescue therapy by 89% [136]. The same research team used roxadustat on patients undergoing dialysis and discovered that it had comparable effects compared with those of parenteral epoetin α [137]. OLYMPUS was a large-scale, double-blinded phase 3 RCT with 2761 participants that evaluated the effects of roxadustat compared with a placebo in patients with NDD-CKD. The key eligibility criteria were CKD stage 3–5, Hgb < 10 g/dL, ferritin ≥ 50 ng/mL, TSAT ≥ 15%, and no ESA use within six weeks. In this trial, roxadustat achieved a statistically significant improvement in Hgb levels compared with the baseline, demonstrating a mean increase of 1.75 g/dL averaged over weeks 28 to 52, whereas this increase was 0.40 g/dL in the placebo group. Compared to the placebo arm, treatment with roxadustat significantly reduced the requirement of blood transfusion by 63%, IV iron by 59%, and ESAs by 87%. Roxadustat also improved Hgb levels compared with the baseline in a subgroup of patients with elevated high-sensitivity C-reactive protein (hsCRP) levels (>5 mg/L), with a statistically significant mean increase of 1.73 g/dL, compared with 0.62 g/dL in the placebo group. Therefore, the OLYMPUS trial concluded that roxadustat effectively increases Hgb levels and that it has more significant beneficial effects in patients with elevated hsCRP levels [138].

The ROCKIES trial compared roxadustat with epoetin α in dialysis-dependent patients with anemia. Roxadustat demonstrated a statistically significant improvement in Hgb level compared with the baseline level, with a mean increase of 0.77 g/dL averaged over weeks 28–52, whereas the mean increase was 0.68 g/dL in individuals administered epoetin α. As a secondary endpoint, roxadustat treatment significantly improved Hgb levels in a subgroup of patients with hsCRP levels > 5 mg/L. In this subgroup of patients, the mean increase in Hgb level was 0.80 g/dL in patients administered roxadustat, whereas it was 0.59 g/dL in individuals administered epoetin α. From week 36 to the end of the study, the patients in the roxadustat group required lower dose IV iron supplementation than those treated with epoetin α (58.7 vs. 91.4 mg per month, respectively; *p* < 0.0001) [139]. Roxadustat is currently approved in China for treating patients with NDD-CKD or ESRD and in Japan for treating patients with ESRD. Roxadustat is also the first orally administered HIF-PH inhibitor to be approved by the FDA for the treatment of anemia in patients with CKD.

At the 2020 American Society of Nephrology Annual Kidney Week, the results of a phase 3 trial for vadadustat (PRO_2_TECT) indicated that it was noninferior to darbepoetin α according to the mean change in Hgb levels compared with the baseline in patients with NDD-CKD [140], but data regarding iron administration has not been addressed. Notably, in the PRO_2_TECT trial, vadadustat did not meet the primary safety endpoint in terms of time to the first occurrence of major adverse CV events (MACEs), defined as a composite of all-cause mortality, nonfatal myocardial infarction, and nonfatal stroke. The results of phase 3 trials evaluating the efficacy and safety of daprodustat for patients with NDD-CKD and those dependent on dialysis are still pending.

### 7.3. Safety Concerns Regarding Hypoxia-Inducible Factor Prolyl Hydroxylase Inhibitors

Because HIFs control many biological processes, undesirable effects may occur in response to the systemic inhibition of HIF-PH. These concerns pertain to a theoretical risk of tumorigenesis, the possibility of increased angiogenesis facilitating the progression of diabetic retinopathy, and adverse CV events.

#### 7.3.1. Tumorigenesis

Tumor cell survival, angiogenesis, and anaerobic metabolism are regulated by HIF. HIF overexpression can be observed in many tumors, and it is related to tumor aggressiveness and distant metastasis [141]. Notably, HIF has been found to promote key steps in tumorigenesis, including angiogenesis, metabolism, proliferation, metastasis, and differentiation. Therefore, whether the drug promotes tumor growth has always been a great concern. In pooled data from the three trials comparing roxadustat with a placebo in patients with NDD-CKD (OLYMPUS, ANDES, and ALPS), a total of 4270 patients were randomly assigned to roxadustat or placebo groups. No clinically meaningful between-group differences were noted in the incidence of neoplasm-related adverse events (AEs). Neoplasm-related AE rates were 2.5 per 100 person exposure years (PEY) in both the roxadustat and placebo groups [142].

In the dialysis trials (ROCKIES, SIERRAS, and HIMALAYAS), 3880 patients were randomly assigned to roxadustat or epoetin α groups. Similarly, no clinically meaningful differences were observed in the organ types of neoplasms, such as prostate or lung cancer or basal cell carcinoma. The neoplasm-related AE rates were 2.7 and 2.3 per 100 PEY in the roxadustat and epoetin α groups, respectively [142].

#### 7.3.2. Angiogenesis

HIF can directly activate the expression of numerous proangiogenic factors, including vascular endothelial growth factor (VEGF), VEGF receptors, plasminogen activator inhibitor-1, angiopoietins, and matrix metalloproteinasis-2 and -9 [143]. Indeed, some reports have suggested that angiogenesis may be associated with the development or progression of retinopathy. Therefore, the mechanism of action of HIF-PH inhibitors has raised concerns regarding whether they could exacerbate pathologic neovascularization associated with diabetic retinopathy, which is common among patients with CKD. In a recently published phase 3 RCT comparing roxadustat with darbepoitein α among patients on HD in Japan, the two groups had a similar incidence of de novo or worsening retinal hemorrhage, and no clinically meaningful changes in retinal thickness were observed. The findings suggest that treatment with roxadustat may not negatively affect retinal neovascularization in patients on HD [144]. However, the observation period of 24 weeks in this study was relatively short. Whether the long-term activation of HIF induces angiogenesis-associated AE should be explored in future clinical trials.

#### 7.3.3. Adverse Cardiovascular Events

HIF-1 plays a critical protective role in the pathophysiology of ischemic heart disease and pressure-overload heart failure [145]. Treatment with HIF-PH inhibitors has the potential to provide a more physiological approach to treating anemia than ESA therapy, which can result in worsening hypertension and increased CV risk during the administration of high doses of ESAs. According to the pooled data from the three trials comparing roxadustat with epoetin α in dialysis-dependent patients (ROCKIES, SIERRAS, and HIMALAYAS), incident dialysis patients receiving roxadustat had a lower incidence of MACEs than patients receiving epoetin α treatment [146]. In pooled data from three trials comparing roxadustat with a placebo in patients with NDD-CKD (OLYMPUS, ANDES, and ALPS), the effects of roxadustat on MACEs were nonsignificant compared with the placebo [142].

According to the findings of the PRO_2_TECT trial, vadadustat treatment was associated with a higher risk of MACEs compared with darbepoetin α treatment in patients with NDD-CKD who had anemia [140]. Although the exact reason for the different results regarding the two HIF-PH inhibitors is unclear, roxadustat has been reported to have potential effects of reducing cholesterol and triglyceride levels as a result of HIF-induced degradation of 3-hydroxy-3-methyl-glutaryl-coenzyme A (HMG-CoA) reductase [147], and this advantage has currently not been reported for vadadustat. Nevertheless, patient safety concerns regarding MACEs after long-term use of HIF-PH inhibitors must be carefully addressed in extended clinical studies and post-marketing investigations.

## 8. Conclusions

Iron deficiency is a common and treatable cause of anemia in patients with CKD. The emergence of new oral and IV iron preparations has widely extended the possibilities of iron supplementation in IDA, especially in the presence of concomitant conditions such as CKD, heart failure, IBD, and cancer. HIF-PH inhibitors are expected to become an alternative treatment for renal anemia. In addition to the benefits of cost reduction and ease of consumption, HIF-PH inhibitors are also expected to be beneficial for treating anemic patients with inflammation and CV diseases. However, all these novel agents for treating IDA in CKD require further investigation in larger patient populations to assess the actual treatment effectiveness and safety. The publication of more clinical trial results is expected to contribute new approaches to renal anemia in patients with CKD.

## Figures and Tables

**Figure 1 ijms-22-01008-f001:**
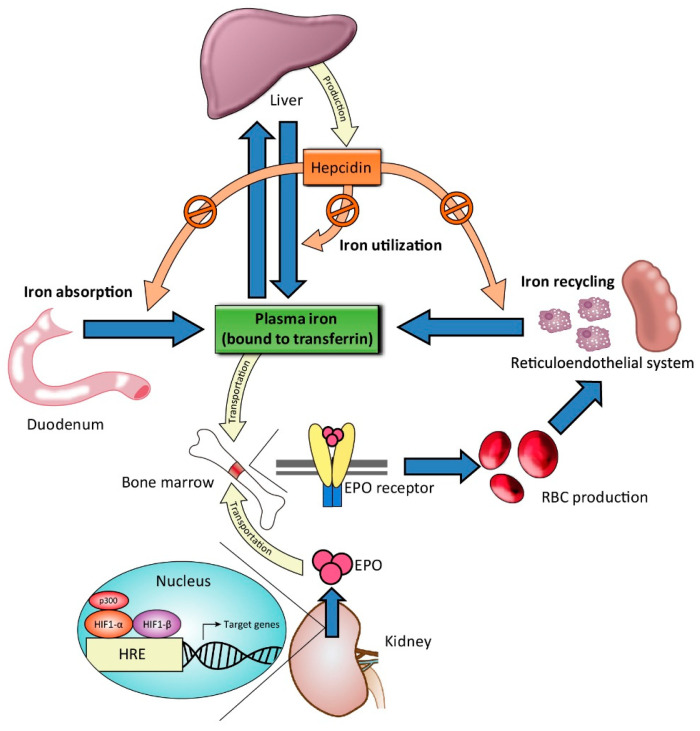
Illustration of iron metabolism. Dietary iron is absorbed from the duodenum. In plasma, iron is bound to transferrin for transport. Some of the iron is stored in the liver through binding to ferritin. The remaining iron is transported to bone marrow for red blood cell (RBC) production, a process stimulated by erythropoietin (EPO), which is secreted by the kidney. EPO transcription is upregulated by hypoxia inducible factors (HIFs), which are transcription factors stabilized under hypoxic conditions. The process involves recruitment of coactivators, such as p300, and binding of HIF and p300 to the hypoxia-response element (HRE) of the EPO gene. RBCs are destroyed at the end of their lifespan by macrophages in the reticuloendothelial system (e.g., in the spleen), and their iron is then recycled. Hepcidin, mainly produced by the liver, plays a central role in the regulation of iron metabolism by downregulating iron absorption and iron utilization under conditions of infection and inflammation.

**Figure 2 ijms-22-01008-f002:**
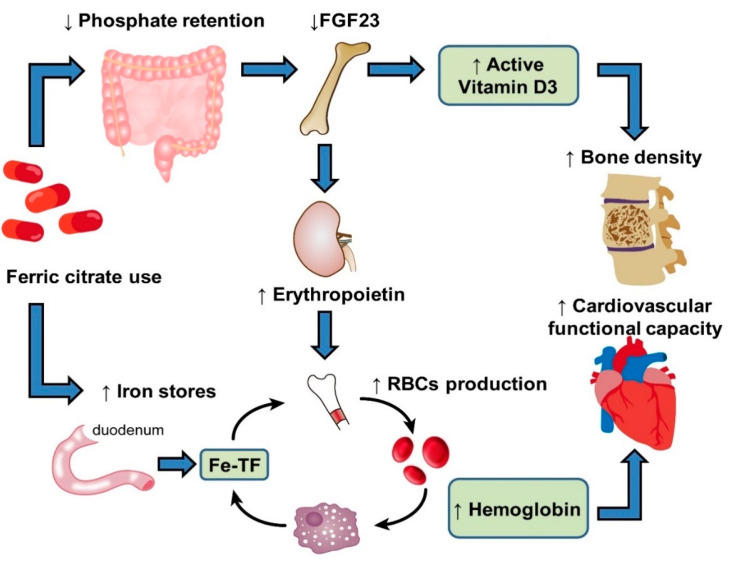
Illustration of the proposed mechanisms of action of ferric citrate (FC). FC binds dietary phosphorus in the gastrointestinal tract, thus controlling serum phosphate levels, which results in reduced fibroblast growth factor 23 (FGF23) production. The resulting reduction of FGF23 levels leads to increased 1,25-dihydroxyvitamin D levels, which subsequently improves bone mineral density. Blocking FGF23 activity also stimulates renal erythropoietin production. FC also enhances intestinal iron absorption and utilization, thereby activating red blood cell (RBC) production in bone marrow and improving hemoglobin (Hgb) levels and cardiac functional capacity. “↑” (means increase), “↓” (means decrease).

**Table 1 ijms-22-01008-t001:** Summary of modern iron therapies in patients with chronic kidney disease.

Drug (Dose Strength)	Common Dosage	Advantages
***Oral iron formulations***		
Ferric citrate (500 mg/capsule)	1000 mg (equivalent to 210 mg iron) three times daily with meals	Phosphate control ↓FGF-23; ↑1,25-dihydroxyvitamin D
Ferric maltol (30 mg of iron/capsule)	1 capsule twice daily before meals	High bioavailability Lipid peroxidation resistance
Heme iron polypeptide (11 mg of iron/tablet)	1 tablet three times daily with meals	Absorption through the intestinal heme transporter
Sucrosomial iron (30 mg of iron/packet)	1 packet once daily after a meal	Unique absorption pathways Good GI tolerance
***IV iron formulations***		
Ferumoxytol (510 mg of iron/vial)	510 mg in a 15-min infusion	Common features: 1. Allows a high-dose IV infusion to quickly be obtained 2. High stability:↓free iron toxicity 3. Low immunogenicity: ↓infusion reactions
Iron isomaltoside 1000 (1000 mg of iron/vial)	1000 mg in a 15-min infusion	
Ferric carboxymaltose (750 mg of iron/vial)	750–1000 mg in a 15-min infusion	
***Intradialytic iron formulations***		
Ferric pyrophosphate citrate (272 mg iron/packet)	1 packet in every 25 gallons of bicarbonate concentrate	Administered through dialysate Iron is transferred to transferrin without iron sequestration

Abbreviations: IV, intravenous; FGF23, fibroblast growth factor 23. “↑” (means increase), “↓” (means decrease).

## Data Availability

Not applicable.

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
