# Peer review of "Iron Therapy in Chronic Kidney Disease: Days of Future Past"

_ijms, 2021, doi:10.3390/ijms22031008_

Round 1
Reviewer 1 Report
This review confers a novelty of new iron therapy to treat renal anemia and would evoke great interests to the patients and doctors. A novel mechanism is raised by the authors.
Some spelling errors should be corrected.
For example, New iron supplement agents have recently 54 emerged, such as ferric citrate (FC), which act as an efficacious and safe phosphate 55 binder that increases iron stores, thereby maintaining Hgb levels.
A spelling error "act" should be "acts"
To make sure the graphs of Figure 1 or Figure 2 is prepared by themself without copy from other figures.
Author Response
Point 1: Some spelling errors should be corrected. For example, New iron supplement agents have recently emerged, such as ferric citrate (FC), which act as an efficacious and safe phosphate binder that increases iron stores, thereby maintaining Hgb levels. A spelling error "act" should be "acts"
Response 1: We have changed “act” to “acts” in Page 2, Para 1, Line 56.
Point 2: To make sure the graphs of Figure 1 or Figure 2 is prepared by themself without copy from other figures.
Response 2: We confirm that two figures in this manuscript are original without copy from other figures.
Reviewer 2 Report
This is a comprehensive, and informative review of iron therapy in CKD. Iron deficiency is a common cause of anemia in patients with CKD. The article is interesting, and it addresses a good clinical topic, however, some points need to be added before publishing.
- The authors may add details on ion channels and transporter in this article.
- Will the presence/absence of other metals influence iron metabolism? Copper is essential for absorbing iron from the gut. When copper levels are low, the body may absorb less iron. Discuss.
- The use of iron chelators (Deroxamine, Deferaxirox, Deferiprone, etc.) will reduce the level of ferritin and iron. Is the use of these drugs associated with CKD?
- Is there any association between iron and vitamins? Vitamin B12 deficiency results in anemia. Discuss.
- The author should consider revising the whole manuscript to make sentences simpler in structure for better reader-friendliness. The authors should proofread and correct the grammar mistake as well as sentence structure.
Author Response
Point 1: The authors may add details on ion channels and transporter in this article.
Response 1: We appreciate your suggestion. We have added details on ion channels and transporter in the Lines 77-85 and Lines 92-95 of Para 2, Page 4 in the revised manuscript.
Line 77-85
Iron from dietary intake is absorbed by the duodenal enterocytes. The duodenal enterocytes reduce dietary ferric iron (Fe3+) to the ferrous state (Fe2+) by ferric reductase, duodenal cytochrome B (Dcytb), and then transport iron into the cell by divalent metal transporter 1 (DMT1). Iron is released into the circulation by export protein ferroportin (FPN) on cell surface. The released of iron to the circulation is coupled to the oxidation of Fe2+ iron to Fe3+ iron, which is carried out by the iron oxidase hephaestin [8]. In the circu-lation, Fe3+ iron binds to plasma transferrin, the major plasma protein that transports iron. Some of the absorbed iron is then transported to the liver, taken up by hepatocyte through transferrin receptors (TfR1 and TfR2) and bound to ferritin for storage.
Line 92-95
High hepcidin concentrations downregulate both duodenal iron absorption and the release of iron from liver and spleen stores by downregulating cell surface FPN expression on absorptive enterocytes, macrophages, and hepatocytes [10].
Point 2: Will the presence/absence of other metals influence iron metabolism? Copper is essential for absorbing iron from the gut. When copper levels are low, the body may absorb less iron. Discuss.
Response 2: We agree with your pertinent comments. In addition to copper deficiency, we have discussed the impact of aluminum overload on iron metabolism in the paragraph “3. Pathophysiology of Renal Anemia”
Line 135-143
3.4. Copper Deficiency
Copper is essential for absorbing iron from the gut. Copper is a crucial component of the ferroxidase enzymes, i.e. hephaestin and ceruloplasmin. Hephaestin is located in the duodenal mucosa and helps in the oxidation of ferrous iron, released from enterocyte FPN, to ferric iron. Ceruloplasmin is essential for iron transfer from macrophage to plasma. Balla and Ismail have reported that hemodialysis (HD) procedure could decrease serum copper levels in HD patients [26]. Therefore, copper deficiency may decrease ferroxidase activity, leading to impairment in iron absorption from the gut and subsequently Hgb synthesis [27].
Line 156-164
3.6. Aluminum Overload
Aluminum had been commonly used as a phosphate binder in patients on dialysis. Nowadays, the use of aluminum in dialysis-dependent patients has mostly been replaced by calcium-containing and non-calcium-containing phosphate binders. However, parenteral aluminum exposure via dialysate contamination is still observed [33]. Aluminum overload could lead to anemia and reduced ESA responsiveness by altered iron metabolism [34], direct inhibition of erythropoiesis [35], and disruption of RBC membrane [36]. Improvement of anemia has been shown in patients using chelation therapy with desferrioxamine (DFO) [37].
Point 3: The use of iron chelators (Deroxamine, Deferaxirox, Deferiprone, etc.) will reduce the level of ferritin and iron. Is the use of these drugs associated with CKD?
Response 3: We are grateful for this comment. We have added the discussion on the iron chelation therapy for iron overload in the paragraph “ 4. Strategies for Iron Management in Patients with Chronic Kidney Disease and End-Stage Renal Disease”
Line 226-256
4.3. Iron Chelation Therapy for Iron Overload
CKD patients are vulnerable to iron overload due to the limited capacity of iron storage and transport via hepcidin modulation. As IV delivered iron exceeds the binding capacity of transferrin, the elevated plasma iron (LPI) contributes to the generation of reactive oxygen radicals that can trigger cellular damage [55]. Apart from the IV iron supplementation, frequent RBC transfusion could contribute to iron overload. The United States Renal Data System (USRDS) data have reported that blood transfusion is still widely used for anemia management in patients with CKD, particularly on dialysis, despite the widespread use of ESAs [56]. Previous literatures have demonstrated that free iron may deposit in various tissues in persistent iron overload states, increasing the risks for liver cirrhosis [57], endocrine dysfunctions [58], diabetes [59], skin hyperpigmentation [60], cardiomyopathy [61], and immune dysfunction [62]. Accordingly, increased awareness of the risk of iron overload in CKD patients requiring chronic blood transfusion or high doses of IV iron is necessary.
Iron chelation therapy has been shown to reduce the iron burden and improve survival in patients with transfusion-dependent anemias [63, 64]. These iron-chelating agents form iron complexes that promote iron excretion, reduce LPI levels, and remove excess iron from cells [65]. Nevertheless, few clinical trials have demonstrated the beneficial effects of iron chelators in CKD or HD patients with iron overload [37, 66]. In clinical practice, the main challenges of iron chelation therapy may result from unfavorable pharmacokinetics and pharmacodynamics. For example, the rapid metabolism (30-minute half-life) of DFO necessitates prolonged IV infusion for 12-15 hours, leading to poor medication adherence. Besides, CKD patients may develop more significant adverse effects such as nephrotoxicity, hepatic failure, agranulocytosis, and teratogenicity in the treatment with deferasirox and deferiprone [67, 68]. Fortunately, a recent experimental study reported a novel DFO-derived nanochelator exerting an improved efficacy and safety in treating mice models of iron overload. The renal clearable nanochelator can be administered subcutaneously and rapidly excreted by urine without demonstrated nephrotoxicity [69]. Compared to native DFO, this new compound may provide a safe and convenient option for iron chelation therapy in CKD patients suffering from iron overload in the future.
Point 4: Is there any association between iron and vitamins? Vitamin B12 deficiency results in anemia. Discuss.
Response 4: We agree with your comments and have added the discussion of vitamin B12 and folate deficiency in the paragraph “3. Pathophysiology of Renal Anemia”
Line 145-154
3.5. Vitamin B12 and Folate Deficiency
Vitamins such as vitamin B12 and folate are necessary for the normal production of RBCs [28]. Many patients on dialysis are taking proton-pump inhibitors, known to be associated with subnormal vitamin B12 levels [29]. Meanwhile, high-flux HD is reported to lower vitamin B12 levels [30]. Vitamin B12 deficiency could cause impairment of folate metabolism by trapping of folate in the form of 5-methyltetrahydrofolate [31], leading to thymidine and purine synthesis impairment, resulting in ineffective erythropoiesis. Dialysis is also associated with a net loss of folate [32], leading to impaired DNA synthesis and ineffective erythropoiesis. However, folate deficiency is typically compensated by a regular diet or routine supplementation of water-soluble vitamins.
Point 5: The author should consider revising the whole manuscript to make sentences simpler in structure for better reader-friendliness. The authors should proofread and correct the grammar mistake as well as sentence structure.
Response 5: We greatly appreciate your constructive comments. Our manuscript has been professionally edited by native English language editors to ensure that the text is optimally phrased and free from typographical and grammatical errors. Please find enclosed the language certificate by professional English language editing company. (Wallace Academic Editing, No. O-2020-010616)
